# Hydrophobic and Luminescent Polydimethylsiloxane PDMS-Y_2_O_3_:Eu^3+^ Coating for Power Enhancement and UV Protection of Si Solar Cells

**DOI:** 10.3390/nano14080674

**Published:** 2024-04-12

**Authors:** Darya Goponenko, Kamila Zhumanova, Sabina Shamarova, Zhuldyz Yelzhanova, Annie Ng, Timur Sh. Atabaev

**Affiliations:** 1Department of Chemistry, School of Sciences and Humanities, Nazarbayev University, Astana 010000, Kazakhstan; darya.goponenko@nu.edu.kz (D.G.); kamila.zhumanova@nu.edu.kz (K.Z.); sabina.shamarova@nu.edu.kz (S.S.); 2Department of Electrical and Computer Engineering, School of Engineering and Digital Sciences, Nazarbayev University, Astana 010000, Kazakhstan; zhuldyz.yelzhanova@nu.edu.kz (Z.Y.); annie.ng@nu.edu.kz (A.N.)

**Keywords:** PDMS, Y_2_O_3_:Eu^3+^, UV protection, hydrophobic coating, solar cells

## Abstract

Solar cells have been developed as a highly efficient source of alternative energy, collecting photons from sunlight and turning them into electricity. On the other hand, ultraviolet (UV) radiation has a substantial impact on solar cells by damaging their active layers and, as a result, lowering their efficiency. Potential solutions include the blocking of UV light (which can reduce the power output of solar cells) or converting UV photons into visible light using down-conversion optical materials. In this work, we propose a novel hydrophobic coating based on a polydimethylsiloxane (PDMS) layer with embedded red emitting Y_2_O_3_:Eu^3+^ (quantum yield = 78.3%) particles for UV radiation screening and conversion purposes. The favorable features of the PDMS-Y_2_O_3_:Eu^3+^ coating were examined using commercially available polycrystalline silicon solar cells, resulting in a notable increase in the power conversion efficiency (PCE) by ~9.23%. The chemical and UV stability of the developed coatings were assessed by exposing them to various chemical conditions and UV irradiation. It was found that the developed coating can endure tough environmental conditions, making it potentially useful as a UV-protective, water-repellent, and efficiency-enhancing coating for solar cells.

## 1. Introduction

Solar cells are playing an important role in global efforts to minimize reliance on conventional energy sources. These technologies not only help to reduce greenhouse gas emissions, but they also provide the potential to boost energy availability in rural or hard-to-reach regions. Despite well-established manufacturing technology, solar panels still face significant challenges that limit their efficiency, stability, and durability. In particular, spectral mismatches between solar cells’ absorbance and solar radiation cause thermalization effects and loss of high energy (UV) and low energy (IR) photons [1,2]. Typically, the maximum conversion efficiency for crystalline silicon solar cells under the AM 1.5 solar spectrum is limited to around 29% [3]. Hence, anti-reflective coatings with various geometry and structure are commonly employed to surpass the Shockley–Queisser limit for single-junction devices [4,5]. On the other hand, another important factor to consider is the exposure of solar cells to UV radiation. Typically, UV radiation is not efficiently absorbed by silicon solar cells and contributes to thermalization and structural degradation processes, resulting in a quick decline in the performance of solar panels over time [6,7,8].

Thin film coatings composed of large-bandgap materials like TiO_2_ and ZnO are commonly suggested to filter UV radiation [9,10]; however, this will result in the cutting-off of UV photons from solar light. To address this issue, down-conversion (DC) optical materials capable of converting UV photons into several visible or infrared photons (quantum cutting effects) can be used [11,12,13,14,15]. Theoretical calculations revealed that a DC layer applied on the front surface of solar cells with Eg = 1.1 eV can boost efficiency by up to 38.6% as compared to 30.9% for cells without a coating [16]. However, under real-world conditions, the efficiency enhancements hardly surpass ~2–3%, which can be attributed to various light photon losses. For example, red-emitting Sr_4_Al_14_O_25_:Mn^4+^,Mg phosphor introduced in the polymethylmethacrylate (PMMA) layer increased the power conversion efficiency (PCE) of perovskite solar cells by ~1.4% [17]. Spin-coated CdSe/CdS quantum dots on the top of c-Si solar cells improved the PCE from 12% to 13.5% [18]. The PCE of commercial c-Si solar cells can also be improved by the liquid-phase deposition of cerium and ytterbium codoped CsPbCl_1.5_Br_1.5_ perovskite DC material [19]. Typically, the average PCE of c-Si solar cells was raised from 18.1% to 21.5%. On the other hand, polyvinyl alcohol PVA-based film containing europium-based ternary complexes shows only a slight PCE improvement by ~0.6% [20]. One can easily observe that in either case, the PCE of the solar cells improves, making this approach practically feasible.

The use of phosphor materials as DC coatings for solar cells can be advantageous due to their excellent chemical stability, quantum yields, and quantum-cutting properties. On the other hand, phosphor materials are vulnerable to humidity, which usually quenches luminescence. Hence, the use of hydrophobic coatings like polydimethylsiloxane (PDMS) has several advantages, including optical transparency, water rejection to prevent the luminescence quenching of phosphor materials, and self-cleaning features [21]. To the best of our knowledge, the use of hydrophobic PDMS with embedded luminescent phosphor particles with high quantum yield to improve the PCE of Si solar cells has not yet been reported. Hence, in this study, we incorporated red-emitting Y_2_O_3_:Eu^3+^ particles into the PDMS matrix to produce a hydrophobic and luminescent coating for solar cells. We found that the developed coating is multifunctional; for example, it can protect the Si solar cells from harmful UV radiation, it has passive radiative cooling, it has self-cleaning properties, and it can also improve the PCE of devices. The hydrophobic properties and structural stability of the produced coating were evaluated by testing it in various types of chemical conditions and under UV irradiation.

## 2. Materials and Methods

### 2.1. Synthesis of Y_2_O_3_:Eu^3+^ Particles

High-purity reagents were purchased from Merck Group (St. Louis, MO, USA) and utilized without any purification. The luminescent Y_2_O_3_:Eu^3+^ particles were produced using the urea homogeneous precipitation protocol [22,23]. In brief, 0.5 g of urea, 371.5 mg of yttrium nitrate hexahydrate, and 12.8 mg of europium nitrate pentahydrate were completely dissolved in 40 mL of deionized water. The resulting mixture was heated in the oven at 90 °C for 2 h. The obtained white precipitates were collected, dried, and calcined in air at 600 °C for 1 h.

### 2.2. PDMS-Y_2_O_3_:Eu^3+^ Coating Deposition

For deposition of the coating with the optimal parameters, 3 mg of the as-prepared Y_2_O_3_:Eu^3+^ powder was dissolved in 25 μL of hydrophobic agent (1H,1H,2H,2H-perfluorooctyltriethoxysilane) and sonicated for 5 min to form a homogenous dispersed mixture. After the addition of 200 μL of cured PDMS (Sylgard 184 kit, Dow Inc., Midland, MI, USA), the solution was thoroughly mixed, and an additional 25 μL of hydrophobic agent was added. The resulting solution was repeatedly sonicated for ~5–7 min. Finally, the obtained solution was spin-coated on glass slides and Si solar cells at 3000 rpm for 20 s. The coated samples were dried for 24 h at 80–100 °C.

### 2.3. Characterization

Morphological and elemental examinations were performed using a scanning electron microscope (SEM, Carl Zeiss Auriga Crossbeam 540, Oberkochen, Germany) equipped with energy-dispersive X-ray spectroscopy (EDX, Aztec Oxford Instruments, Abingdon, UK). The photoluminescence analysis, quantum yield, reflectance and absorbance measurements were carried out with a Quantaurus absolute quantum yield spectrometer (C9920-02, Hamamatsu Photonics K.K., Hamamatsu, Japan) equipped with an integrating sphere. X-ray diffraction (XRD) measurements were performed using a SmartLab X-ray Diffractometer (Rigaku Corp., Tokyo, Japan) with a Cu Kα radiation source. Transmittance measurements were performed using the Genesys 50 UV–Visible spectrophotometer (Thermo Fisher Scientific Inc., Waltham, MA, USA). The hydrophobic properties of the films were tested by contact angle goniometer (Ossila Ltd., Sheffield, UK). The current–voltage (J-V) measurements were performed using a semiconductor parameter analyzer (Agilent B1500A, Agilent Technologies Inc., Santa Clara, CA, USA). The samples were illuminated using AAA class Oriel Sol3A solar simulator (Newport-Spectra Physics GmbH, Darmstadt, Germany). An AM 1.5 G filter and Si reference cell were applied to adjust the light intensity.

## 3. Results and Discussion

The deposition process was optimized through testing of several concentrations of Y_2_O_3_:Eu^3+^ particles in PDMS and also by varying the thickness of the coatings. However, the experimental data discussion is confined to the optimal coating conditions only. Figure 1 displays the overall scheme and processes that take place when light photons with different energies hit the surface of bare glass or glass with a coating. Some photons will be reflected, while others will pass through and generate electron–hole pairs in the active layer of the solar cell. Down-conversion optical materials have absorption in the UV-blue range and are commonly deposited on top of solar cells, which in turn simplifies the coating process. In this study, the PDMS acts as a hydrophobic matrix that houses the downconversion particles, while these particles convert UV-blue photons into visible light photons. In this study, Y_2_O_3_:Eu^3+^ particles were selected as a down-conversion optical material because of the synthesis simplicity, large Stokes shift, and high quantum yield [22].

The morphological examination of the Y_2_O_3_:Eu^3+^ particles was conducted using TEM and SEM. Figure 2A shows that the produced particles have a spherical shape and range in size from ~350 to 500 nm. Appendix A confirms the even distribution of the key elements in the sample, with yttrium (Y), oxygen (O), and europium (Eu) effectively detected. Figure 2B shows a cross-sectional image of the hydrophobic and luminescent coating taken for the optimal sample. Cross-sectional SEM analysis revealed that the coating thickness was ~5.4 μm for the optimal samples. Furthermore, the successful incorporation of Y_2_O_3_:Eu^3+^ particles into a PDMS matrix was also validated, with some Y_2_O_3_:Eu^3+^ particles visible at the border of the coating. The XRD analysis of the prepared Y_2_O_3_:Eu^3+^ particles (Appendix A revealed distinct diffraction peaks at 2θ angles of 20.7, 29.2, 33.8, 35.9, 39.9, 43.5, 48.6, and 57.6, corresponding to the diffraction of the (211), (222), (400), (411), (332), (413), (440), and (622) crystal planes, respectively. The observed XRD pattern suggested that the Y_2_O_3_:Eu^3+^ particles adopted a body-centered cubic (bcc) phase of Y_2_O_3_ [24].

Figure 3A shows the “Commission Internationale de l’éclairage” CIE chromaticity diagram and excitation/emission spectra of Y_2_O_3_:Eu^3+^ particles. The emission pattern of Y_2_O_3_:Eu^3+^ particles is represented by ^5^D_0_→^7^F_0_, ^5^D_0_→^7^F_1_, ^5^D_0_→^7^F_2_, ^5^D_0_→^7^F_3_ transitions in the yellow–red region, with the ^5^D_0_→^7^F_2_ electric dipole transition at 612 nm being the most pronounced and dominant. The corresponding transitions are schematically shown in Appendix A. The excitation curve (λ_em_ = 612 nm) shows a broad band in the UV region, which is associated with a charge transfer band from the 2p orbital of O^2−^ to the 4f orbital of Eu^3+^ [22,24]. The measured absolute quantum yield QY of Y_2_O_3_:Eu^3+^ particles was found to be ~78.3%, which is considered to be exceptionally high compared to other organic/inorganic optical materials [25,26]. The CIE diagram also confirmed the successful light conversion from UV to red, with the following emission chromaticity coordinates (x = 0.591; y = 0.330). It should be outlined that the majority of solar cells, including silicon, perovskite, and dye-sensitized have notable light absorption in the visible–near IR regions. Hence, UV to red down-conversion of Y_2_O_3_:Eu^3+^ particles with high QY can be used to shield solar cells from destructive UV radiation and at the same time improve the efficiency of solar cells by supplying additional photons. In the next step, we tested the light transmittance of the PDMS-Y_2_O_3_:Eu^3+^ coating with optimal thickness to that of the bare glass slide. Figure 3B shows a small reduction in light transmittance when compared to the reference glass slide. In particular, the light transmittance was reduced by 12.7% (at 320 nm), 11.5% (at 500 nm), and 8.5% (at 800 nm) after deposition of the PDMS-Y_2_O_3_:Eu^3+^ coating. Figure 3B inset shows that the PDMS- PDMS-Y_2_O_3_:Eu^3+^ coating with average light transmittance of ~80% or more was visually transparent in white light but glowed red when exposed to UV light.

Figure 4 depicts the absorbance and reflectance studies of the PDMS-Y_2_O_3_:Eu^3+^ coating measured in an integrating sphere. One can notice that absorbance of the PDMS-Y_2_O_3_:Eu^3+^ coating was observed in the UV range (~260–400 nm) with a maximum close to 304 nm. Hence, we can speculate that the PDMS-Y_2_O_3_:Eu^3+^ coating indeed only absorbed UV photons and converted them further into visible light photons. A similar trend was also observed with the reflectance study; a minimal reflection was observed in the region of ~260–400 nm, which corroborated well the absorbance results.

The hydrophobicity of the formed PDMS-Y_2_O_3_:Eu^3+^ coatings was assessed by measuring the contact angles using a goniometer. The optimized samples displayed contact angles ranging from 119 to 121°, indicating their hydrophobic properties. Furthermore, the structural stability of the coating was further assessed by a series of additional experiments. For example, the UV stability was estimated by placing the coated glass slides under an ultraviolet lamp (λ = 365 nm, 8 W) at a distance of 2 cm for 24 h, with a measuring of the contact angle every 3 h. Moreover, the chemical stability of the coating was determined by immersing the coatings in different chemical environments, i.e., pH 3, pH 5, pH 7 (distilled water), and NaCl (1M) solutions for 24 h. Figure 5A shows that after being exposed to UV light, the measured contact angles ranged between 118 and 120°, indicating good resistance of the coating to UV irradiation. It should be outlined that the UV flux reaching the Earth’s surface is lower than that of a UV lamp; hence, the developed PDMS-Y_2_O_3_:Eu^3+^ coatings have the potential to endure long-term sun irradiation. Figure 5B indicates that the contact angles of the coatings also did not vary substantially after being placed in various chemical media simulating acidic conditions. Hence, it can be concluded that the PDMS-Y_2_O_3_:Eu^3+^ coating may retain hydrophobic properties for an extended period, which is potentially useful for self-cleaning purposes.

As a proof-of-concept, the photovoltaic (PV) parameters of polycrystalline Si cells (active area = 1 × 1 cm) were tested before and after the deposition of PDMS-Y_2_O_3_:Eu^3+^ with the optimized thickness. Figure 6A,B illustrate the corresponding J-V curves and external quantum efficiency (EQE) of the Si cells (n = 3) before and after the deposition of the coating, respectively. Figure 7 shows that all key PV parameters were improved; i.e., one can achieve PCE improvement in the Si cells following PDMS-Y_2_O_3_:Eu^3+^ deposition. On average, the PCE enhancement was found to be ~9.23%. It should be also emphasized that the EQE pattern of the coated sample followed the proposed idea, i.e., to absorb photons in the UV-blue region and convert them to visible photons in the red region. Generally, all key PV changes can be associated with the management of the optical characteristics, and a similar trend was observed in the literature [27,28]. Typically, one can observe that the V_oc_, J_sc_, and FF values were improved to ~3.57%, 1.38%, and 10.1% respectively. Among them, the J_sc_ enhancement was associated with increased light-generated current, while the V_oc_ improvement was typically associated with improved concentration or lower recombination rates of charge carriers [29,30]. The FF is frequently linked to parasitic resistive losses [31], and the FF can gradually increase with the growth in the irradiance (in this case, irradiance is increasing in the red and near IR regions), which reaches a maximum point and then decreases [32]. Furthermore, the FF lowers, as the temperature rises [32]; so, the PDMS-Y_2_O_3_:Eu^3+^ coating also reduces the heating of Si cells similar to a PDMS-SiO_2_-based radiative cooling film [33]. For example, under solar illumination, an uncoated Si cell reached approximately 54.9 °C in 5 min, whereas a coated cell reached approximately 52.7 °C, as shown in Appendix A. Hence, we can speculate that the overall PCE enhancement is associated with the synergetic combination of light conversion by PDMS-Y_2_O_3_:Eu^3+^, improved generation of charge carriers in Si cells, and passive cooling of the PDMS-Y_2_O_3_:Eu^3+^ coating.

The self-cleaning property of the prepared PDMS-Y_2_O_3_:Eu^3+^ coating was further shown on a glass slide (not visible on the Si cells due to the dark surface). Figure 8 shows the digital images of the coated glass slide with soil spots on the surface. Because the adhesion between the surface and dust is weaker than that between the water droplet and dust, the water droplet will clean the surface coating as it rolls down.

Finally, the UV protection capabilities of the coatings with optimized thicknesses were investigated under the constant UV lamp illumination (λ = 365 nm, 8 W, t = 1 h) of the bare and coated Si cells (n = 3 per batch). We found that, on average, the coated Si cells had a lower PCE drop (~0.8%) as compared to those of the bare Si cells (~1.8%). Hence, it can be concluded that PDMS-Y_2_O_3_:Eu^3+^ can be potentially employed as a hydrophobic UV-blocking and light-converting coating for the potential power enhancement of Si cells. Si solar cell fabrication is a mature process, with a reported efficiency degradation by several % within a year [34]. As a result, the short-term testing of non-coated and coated Si cells at 80 °C yielded no appreciable decrease in the PCE. Therefore, field trials, such as film deposition using spray coating on large-sized panels, as well as long-term durability testing under temperature, humidity, and light fluctuations will be explored in future studies.

## 4. Conclusions

In summary, we successfully prepared a hydrophobic and luminescent PDMS-Y_2_O_3_:Eu^3+^ coating for potential applications in photovoltaic solar cells. The prepared coating contains Y_2_O_3_:Eu^3+^ particles, dispersed in a hydrophobic PDMS matrix, which convert UV photons to visible (red) photons with a QY of ~78.3%. As a result, down-converted red photons can be partially reabsorbed by Si cells, resulting in a PCE improvement of ~9.23%. The preliminary data on the UV stability, chemical stability, and UV protection suggested that the PDMS-Y_2_O_3_:Eu^3+^ coating is durable and can be employed on UV-sensitive solar cells for UV protection and power enhancement purposes.

## Figures and Tables

**Figure 1 nanomaterials-14-00674-f001:**
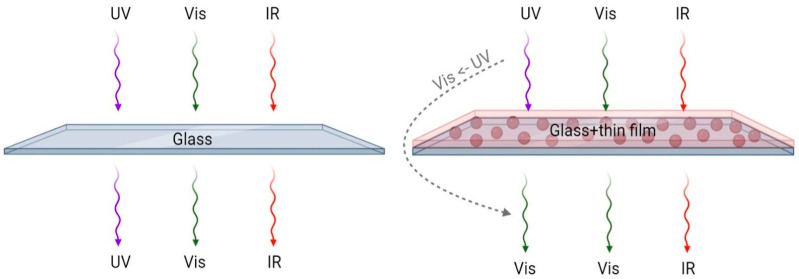
Schematic representation of light conversion by photoluminescent coating.

**Figure 2 nanomaterials-14-00674-f002:**
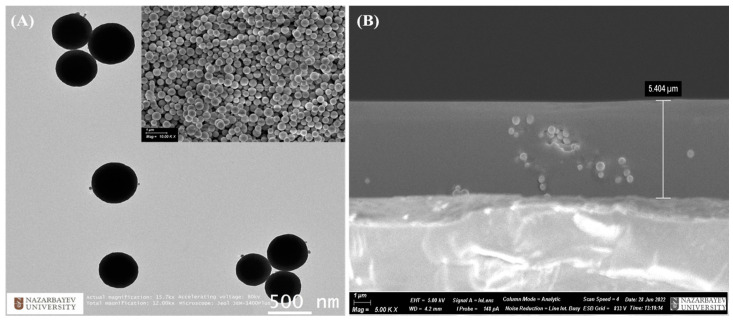
(**A**) TEM and SEM (inset) images of Y_2_O_3_:Eu^3+^ particles. (**B**) Cross-sectional SEM image of PDMS-Y_2_O_3_:Eu^3+^ coating.

**Figure 3 nanomaterials-14-00674-f003:**
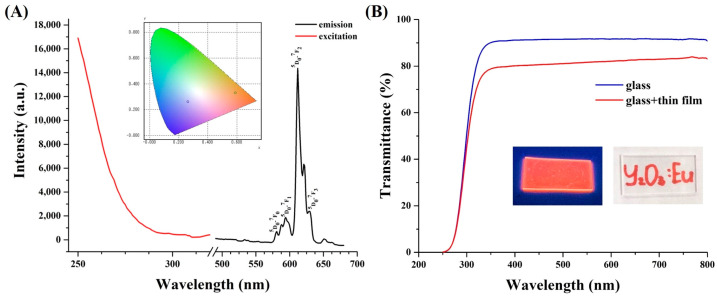
(**A**) PL emission and excitation of Y_2_O_3_:Eu^3+^ particles. (**B**) Transmittance of a bare glass slide and a glass slide with a PDMS-Y_2_O_3_:Eu^3+^ coating. Inset are digital images of the PDMS-Y_2_O_3_:Eu^3+^ coating under UV and daylight illumination.

**Figure 4 nanomaterials-14-00674-f004:**
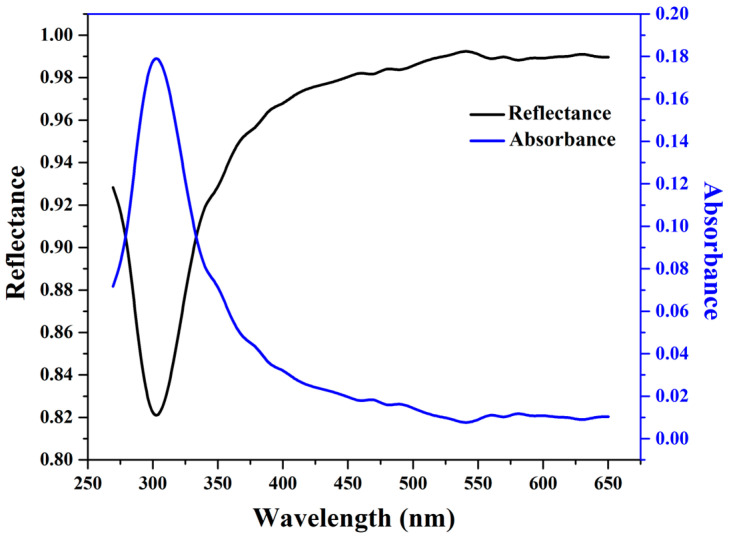
Absorbance and reflectance measurements of PDMS-Y_2_O_3_:Eu^3+^ coating on glass slide.

**Figure 5 nanomaterials-14-00674-f005:**
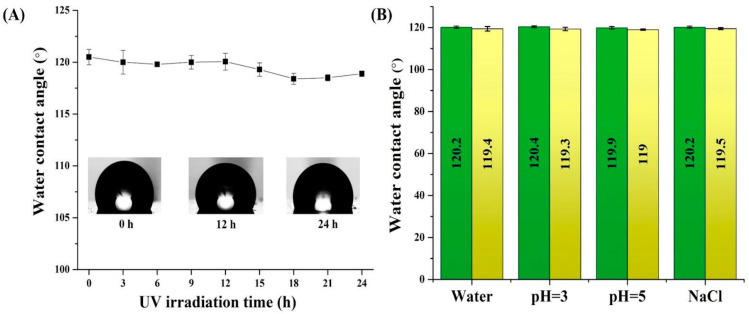
Effect of (**A**) UV irradiation time and (**B**) impact of various media on the hydrophobicity of PDMS-Y_2_O_3_:Eu^3+^ coatings.

**Figure 6 nanomaterials-14-00674-f006:**
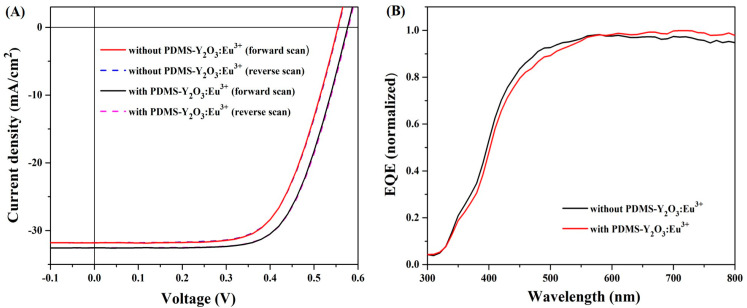
(**A**) J–V and (**B**) EQE characteristics of Si solar cells with and without PDMS-Y_2_O_3_:Eu^3+^ coatings.

**Figure 7 nanomaterials-14-00674-f007:**
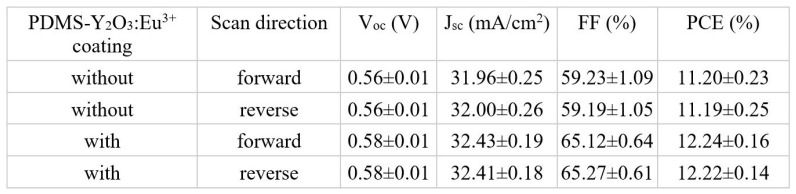
Key PV parameters for Si cells before and after the deposition of luminescent PDMS-Y_2_O_3_:Eu^3+^ coatings.

**Figure 8 nanomaterials-14-00674-f008:**
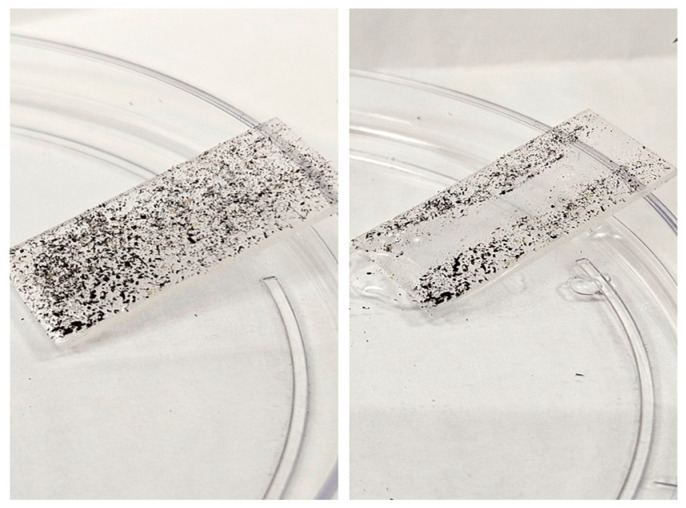
Demonstration of the self-cleaning properties of the PDMS-Y_2_O_3_:Eu^3+^ coating. The right image depicts the coating after several water droplets have been dropped.

## Data Availability

The original contributions presented in the study are included in the article/Appendix A; further inquiries can be directed to the corresponding author.

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
