# Peer review of "Hydrophobic and Luminescent Polydimethylsiloxane PDMS-Y2O3:Eu3+ Coating for Power Enhancement and UV Protection of Si Solar Cells"

_nanomaterials, 2024, doi:10.3390/nano14080674_

Round 1

Reviewer 1 Report

Comments and Suggestions for Authors

This paper reports a hydrophobic and luminescent coating using polydimethylsiloxane (PDMS) combined with red-emitting Y2O3:Eu3+ particles for photovoltaic applications. This coating offers the protection of silicon solar cells against ultraviolet (UV) radiation harm, while simultaneously boosting their power generation by converting UV photons into visible red photons. The work is interesting, and the paper is well articulated. However, there are several crucial issues that should be further addressed in the manuscript.

1. Please clarify the novelty of this work. There have been reports on the efficiency enhancement of solar cells using downconversion or upconversion nanoparticles at the surface of devices [Du et al., Nanoscale Research Letters 10, 321 (2015); Lee et al., Advanced Functional Materials, 32, 2204328 (2022); Sekar et al., ACS Omega, 7, 35351 (2022)].

2. Can the authors provide the element information of luminescent particles (Y2O3Eu3+) in Figure 2, such as an EDX map or spectrum?

3. Can the authors provide the energy level diagram of downconversion luminescent particles (Y2O3Eu3+)?

4. Nanoparticles can cause light to scatter or diffuse. Therefore, please further provide the optical properties of glass+thin film (PDMS-Y2O3:Eu3+ coating), including total transmittance, diffuse transmittance, reflectance, and absorption (1-reflectance-total transmittance).

5. For the solar cell performance,

1) Please provide the current-voltage characteristic curves under solar light illumination, including performance parameters of current density, voltage, fill factor, and power conversion efficiency.

2) To verify the downconversion effect of Y2O3:Eu3+ particles, the incident photon to current conversion efficiency (IPCE) spectra should be provided in the manuscript.

3) For applications in harsh outdoor environments, the durability, thermal stability, and long-term stability should be tested on the solar cell performance.

6. Can the authors demonstrate the self-cleaning effect (e.g., dust removal by rolling droplets) of the proposed coating on the solar cell? 

Comments on the Quality of English Language

The paper is generally well-written with good English grammar and clarity of expression.

Author Response

A separate file is attached! 

Reviewer 2 Report

Comments and Suggestions for Authors

This manuscirpt reported that the Y2O3:Eu@PDMS films was emploied as the the light conversion layer to enhance the the photoelectric conversion efficiency of the Si solar cells.  This work is innovative and systematic. Thus, I recommend it can be published in nanomaterials after minor revision.

1. How about the external quantum efficiency of the Y2O3:Eu@PDMS films with an excitation wavelength in the range rom 200-300 nm?

2. The operational stability of the Y2O3:Eu@PDMS film-based of the Si solar cells should be investigaged.

3. Some related reference should be cited, such as, RSC Adv. 2015, 5, 102682-102688; RSC Adv. 2015, 5, 7673-7678.

Comments on the Quality of English Language

 Minor editing of English language required

Author Response

A separate file is attached! 

Round 2

Reviewer 1 Report

Comments and Suggestions for Authors

The manuscript has been revised based on my suggestions and comments. 

1. However, there are still practical application issues to address, such as durability, thermal stability, and long-term stability. 

2. Additionally, the reasons behind the 9.8% improvement in the PCE of the solar cell with PDMS-Y2O3:Eu3+ coating compared to one without the coating should be discussed in more detail. Specifically, the increase in Voc and FF by approximately 3.5% and 10%, respectively, while Jsc only increases by about 1.5%, warrants further discussion. Given that the PDMS-Y2O3:Eu3+ coating absorbs UV-Blue light and converts it to red light, it primarily affects the photocurrent of solar cells. Therefore, the authors should provide a detailed discussion on this aspect.

Comments on the Quality of English Language

Minor editing of English language required

Author Response

A pdf file is attached. 
